# Visna—Visualising Multivariate Missing Values

**Antony Unwin** [* 1]   **Alexander Pilhoefer** [*]

## Abstract

Identifying patterns of missing values in large datasets is difficult. The visna plot is a matrix visualisation with variables in the columns and patterns of missing values in the rows. Both rows and columns can be sorted and filtered. Visna is a powerful graphical tool for exploratory analysis.

## 1. Introduction

The R Task View for Missing Data (Josse et al., 2020) provides a good discussion of the issues involved in dealing with missing values. The first stage they define is 'Exploration of missing data' and visualisation plays an important role. Several graphical tools are available for investigating datasets, especially for small numbers of variables and small numbers of cases. More tools are needed for larger datasets, such as those that arise in Machine Learning. Datasets may be large because they have many cases (when displays with one row per case do not work), because they have many variables (when displays without filtering of variables do not work), or because they include many missing values (when all displays may have difficulties). There is an extensive research literature on methods for dealing with missing values and many models for imputation have been suggested. These depend to a great extent on the patterns of missingness in the data (Molenberghs et al., 2014). Graphical overviews help to identify and study these patterns.

There have been interactive graphics software packages that incorporated missing value displays, in particular MANET (Hofmann, 2000) and Mondrian (Theus, 2005). Both allowed linking across multiple windows so that subsets of the data, for instance groups of missing values, could be investigated across all the variables in a dataset. Both were standalone packages, mainly for graphics. Should interactive graphics tools be fully implemented in R, then these features in MANET and Mondrian would be valuable for missing value analyses.

Several R packages offer case by variable matrix visualisations for displaying missing values of a dataset (e.g., *VIM* and *visdat*) and offer some sorting. *VIM* also offers a visualisation of combinations by variables including an interesting option for colouring imputed data. Visna plots complement these by dealing with larger datasets and offering more sorting methods. They were introduced originally in (Malik, 2010a) and the *vmv* package (Malik, 2010b), were discussed in (Unwin, 2015), and have been implemented more fully in the R package *extracat* (Pilhoefer & Unwin, 2013).

Visna plots use matrix visualisations showing missing value patterns by variables. Cases have the same pattern of missing values if they are missing on the same group of variables. Figure 1 shows an example of a visna plot for the ocean-buoys dataset in the *naniar* package. The rows and columns have been sorted by the marginal numbers. Some visualisations of missing value patterns were discussed in (Fernstad, 2019). There has been a lot of work on matrix visualisation in general (Wu et al., 2008) for a variety of applications. A key feature is the ability to sort the rows and columns in multiple ways to highlight different structures in the data.

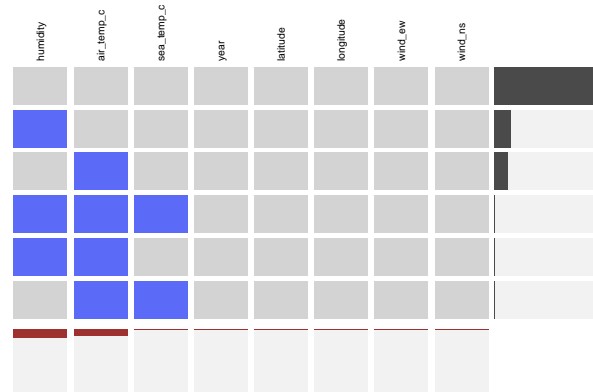

*Figure 1.* Visna plot of the 8 variables and 736 cases in the ocean-buoys dataset. The majority of cases have no missings. The two main missing patterns arise when either humidity or air temperature are the only values missing.

---

[*]Equal contribution  [1]Institute of Mathematics, University of Augsburg, Augsburg, Germany. Correspondence to: Antony Unwin <unwin@math.uni-augsburg.de>.

*Presented at the first Workshop on the Art of Learning with Missing Values (Artemiss) hosted by the $37^{th}$ International Conference on Machine Learning (ICML).* Copyright 2020 by the author(s).

## 2. What do you want to know?

Missing values can be mildly irritating (if there are only a very few of them), straightforward but requiring effort (if there are a larger number missing at random in a known way), or a problem (if there are many that are not missing at random). Before analysing a dataset it is good to know what the situation is overall and in detail. There are some of the many questions to be answered.

- Are there missing values?

- If so, how many missings are there? What proportion of the data?

- Are there missings in all of the variables or just some?

- How many are missing in each variable? What are the proportions of missings?

- How many complete cases are there (cases with no missings)?

- How many values are missing for each case? What proportions are missing?

- What patterns of missing values are most frequent? If cases have the same pattern, is there anything else they have in common?

- Are there patterns of missings amongst variables? Perhaps there are variables that are always missing together, perhaps some only have missings if others do. Does it depend on the type or subject matter of the variable?

## 3. Visna plots

A visna plot is a matrix visualisation of the missing values in a dataset. It assumes that missing values are coded as NA and has one column for each variable in the dataset and one row for each missing value pattern. Additionally there are barcharts in the bottom and right margins showing the proportions of missing values per variable (column margins) and the numbers of the different missing value patterns (row margins).

Two cases have the same missing value pattern if they are missing on the same variables and not missing on any others. If a dataset has p variables (is p-dimensional) then there are $2^p$ possible missing value patterns, including the patterns of no missings and all missing. This number can get very big very quickly as $p$ increases, but the actual number of missing value patterns for any dataset will often be quite low and, at any rate, must always be less than or equal to $n$ the number of cases in the dataset. For instance, there are 8 variables in the oceanbuoys dataset in Figure 1 . Potentially

there could be $2^8 = 256$ patterns, but only 3 variables have missings and there are only 6 missing value patterns.

Figure 2, a visna plot of data from the gene dataset of the *missMDA* package , shows that even when there are many variables (over 400 here), some structure may still be visible. To preserve the original order, the columns have not been sorted. The first group of columns represents CGH data and you can see that only one case has any missing values for these, four in all. The second group of columns showing a set of cases with the same large number of missing values represents gene expression data.

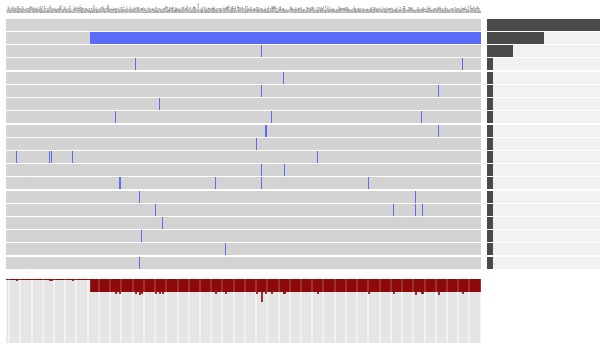

*Figure 2.* Visna plot of the 432 continuous variables and 54 cases in the gene dataset. One group of cases has many missings, otherwise there are few missings.

## 4. Sorting and filtering

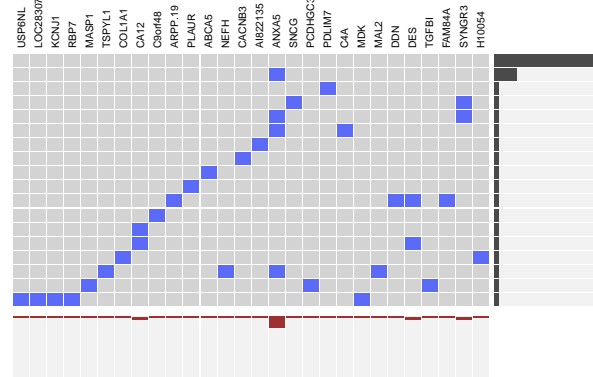

*Figure 3.* Visna plot of the gene dataset with the 11 cases with many missings removed and only variables with some missing values for the remaining cases shown. The rows have been sorted but not the columns.

Sorting rows is important for emphasising which patterns arise most often (and showing which arise very rarely). If the

case data have been collected in a particular order, perhaps patients one after another, it could be useful to preserve that order, but usually the order of cases in a dataset is fairly arbitrary. Sorting columns is useful for highlighting the variables with the most missings. There may be a natural order to the variables, for instance if they are values for successive years, and then it is best to leave them in their default order. As with the gene data, where there were two groups of variables, it can be sensible to keep the groups together. Other sortings may be considered amongst the many available for matrix visualisations, but sorting by rows, columns or both is often very informative.

Filtering can be used to show only the variables with the most missings and/or the patterns arising most often. When there are many variables you do not want to try to show them all. You could either specify the number of variables to be shown or the percentage of all missings to be covered by the variables to be shown (the parameter $f_c$ in *extracat*'s visna function is used for this). The same can be done for the rows if you do not want to show rows for patterns which rarely arise (the parameter $f_r$ is used for the rows). Only the $f_r$ most frequent rows are kept if $f_r > 1$. Values of $f_r < 1$ are interpreted as proportions and the minimum number of rows covering at least $f_r$ percent of the observations are kept.

The more variables there are and the more cases there are, the more likely it is that there are more rare missing value patterns. An example of sorting and filtering is shown in Figure 3. Filtering out the cases with no expression data and the variables with no missings on the remaining cases makes the display much more readable.

## 5. Application to a large survey dataset

The riskfactors dataset in the R package *naniar* is a subset of 245 cases and 34 variables from the Behavioral Risk Factor Surveillance System (BRFSS) Survey of 2009 (CDC, 2009). Figure 4 is a visna plot of the data. Some of the information that can be seen is listed in the caption. It is initially surprising that there are hardly any missings on age, but the survey code book says that people could refuse (coded 9) or not know/not be sure (coded 7). It turns out these two answer alternatives were available for many questions, so a full study of missing values should include a discussion with the CDC to agree what should be treated as missing for any given analysis.

As the full dataset is available online, it was downloaded and a visna plot Figure 5 drawn of the same 34 variables as in Figure 4 for all 432607 cases. (Actually, only 33 variables were used because BMI is a calculated variable. In addition the codings for weight and height are complicated as they allow either English or metric units and this is part of the

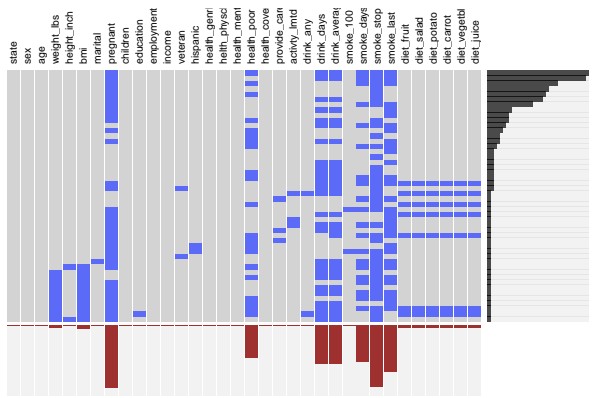

*Figure 4.* Visna plot of the riskfactors dataset. The rows have been sorted but not the columns. Amongst other things that can be seen, the pregnancy question was not asked of men or older women, so there are many missings; BMI was calculated from weight and height, so has to be missing when one or both are missing; a few respondents answered none of the diet questions, the others answered them all. There are no complete cases.

coding.)

Comparing Figure 4 and Figure 5 you can see that the proportions of missing values by variable (the bottom barcharts) are fairly similar. You can also see that there are far more missing value patterns (more rows) in Figure 5, as you would expect with a much larger dataset. It is still true for the larger dataset that most respondents answered all the diet questions (the block to the far right), but it is no longer true that the rest answered none of them. Closer inspection (and checking with code to be absolutely sure!) reveals that there are no complete cases in this dataset.

If we restrict ourselves to the missing value patterns that together include at least 99% of all the cases, we get Figure 6, which has a lot of similarity with Figure 4.

## 6. Conclusions

Missing values may influence analyses. They can be ignored or values can be imputed for them in one of a number of ways. To help decide how best to deal with them, it is important to know how many missing values there are, whether there is a structure to the missingness, and, if so, what kind of structure. Visna plots are an effective way of getting a multivariate overview of the missing values in a dataset.

### Software and Data

The graphics in this paper were drawn using R and the package *extracat*. The package is available on github (https://github.com/heike/extracat). Numbers of cases do

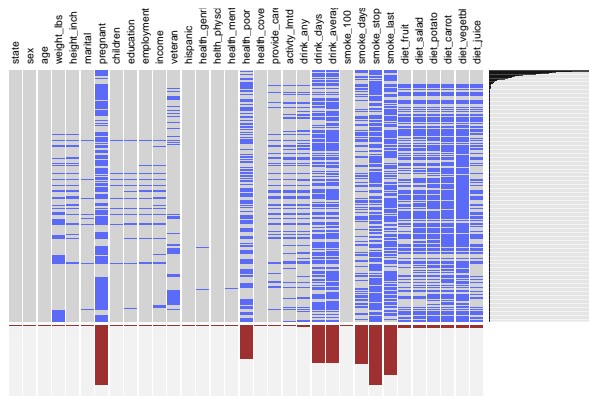

*Figure 5.* Visna plot for all participants of the BRFSS survey of 2009 for (almost) the same variables as the previous plot. The rows have been sorted but not the columns.

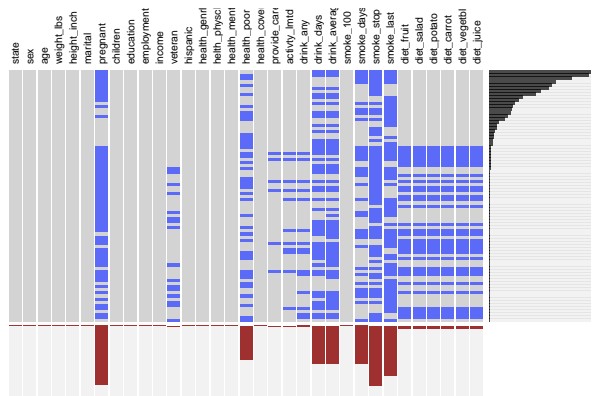

*Figure 6.* Visna plot for the BRFSS survey restricted to the patterns covering 99% of the missings.

not have much effect on speed, but numbers of variables do. The datasets used are available in R packages, apart from the BRFSS one that can be downloaded from the web.

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
