# OpenReview forum: "Visna---Visualising Multivariate Missing Values"
_ICML.cc/2020/Workshop/Artemiss — ICML Artemiss 2020_

### Official Review · AnonReviewer2 · 2020-06-22
**Review 'Visna – Visualising Multivariate Missing Values'**

**Rating:** 8
**Confidence:** 5

**Review:**

The workshop submission addresses the visualization of multivariate missing values in large datasets (many cases, many variables). The authors provide a review about matrix plots / visna plots and specifically talk about challenges that come along while applying these to large datasets.

RELEVANCE:
The submission is about the application of missing data visualization techniques. Thus, it is very relevant for the workshop on missing data.

SIGNIFICANCE OF THE PROBLEM:
Missing data can occur nearly everywhere, where data gets recorded. The more complex and extensive the created dataset gets, the higher is usually the chance of having some cases of missing values. Therefore, this is an important problem in real-world applications.

ORIGINALITY OF THE WORK:
The submitted work does not necessarily advance the state-of-the art, but it gives an interesting short sum-up about current techniques. Which is totally fine as per the workshop acceptance criteria.

WRITING QUALITY:
Language and grammar of the submitted short paper are very good. The figures are of high quality. It is publication ready in this state.

TECHNICAL SOUNDNESS:
The authors show good knowledge of their topic and their explanations and descriptions are scientifically correct.

REPLICABILITY:
The required tools/packages are linked and corresponding literature is cited. Would have been nice, if the code for creating the plot examples from the paper were also available.

---

### Official Review · AnonReviewer1 · 2020-06-23
**Interesting tool for the representation of large dataset with matrix visualization**

**Confidence:** 4
**Rating:** 7

**Review:**

# SUMMARY:

The authors present a graphic tool to plot datasets, highlighting the missing values for each variable, using matrix visualizations. The possibility of sorting and filtering rows and columns is included. It is shown that when the dataset became larger more missing value patterns arise (more rows in the resulting matrix). The main goal is to identify possible patterns of missing values in the dataset.

# PROS:
- The paper is totally in line with the topic of the workshop.
- There are applications with datasets with different sizes.
- The images have a good quality.

# CONS:
 - For a better understanding of the applications, a brief explanation of the datesets should be considered.
 - From the originality point of view, even if it propose a usefull tool, the paper does not increase the knowledge in the domain of missing values.
 - The paragraph _"4. Sorting and filtering"_ lacks of clarity: how the parameters  $f_r$ and $f_c$ have been defined? It is not clear to me what does it happen when $f_r <1$.
 - In the conclusions it is mentioned the speed of the code, it would be interesting to investigate how the number of variables affects the speed of the this method.

## Minor remarks and suggestions:
- Sometimes the syntax tends to be too heavy. For example, I think there are too many brackets in paragraph  _"1. Introduction"_ and at the beginning of  paragraph _"2. What do you want to know?"_. In this way the reading flows slowly and it is difficult to follow the author.
- I would suggest to explain or rename the variables in _Figure 3_, it is difficult to interpret the plot without knowing to what the variable are referrring to.
- The authors could underline in a better way what is their additional contribution with respect to the R package _"extracat"_.

---

### Decision · Program_Chairs · 2020-07-02

**Decision:**

Accept

**Comment:**

We're happy to accept this paper at Artemiss. We'll contact you soon to inform you about more details concerning the format of your presentation at the workshop, and the camera-ready version deadline. Please take into account the referee's comments to write the camera-ready version.